# Angiotensin Converting Enzyme Inhibitors May Increase While Active Vitamin D May Decrease the Risk of Severe Pneumonia in SARS-CoV-2 Infected Patients with Chronic Kidney Disease on Maintenance Hemodialysis

**DOI:** 10.3390/v14030451

**Published:** 2022-02-22

**Authors:** Piotr Tylicki, Karolina Polewska, Aleksander Och, Anna Susmarska, Ewelina Puchalska-Reglińska, Aleksandra Parczewska, Bogdan Biedunkiewicz, Krzysztof Szabat, Marcin Renke, Leszek Tylicki, Alicja Dębska-Ślizień

**Affiliations:** 1Department of Nephrology Transplantology and Internal Medicine, Medical University of Gdańsk, 80-210 Gdańsk, Poland; ptylicki@gumed.edu.pl (P.T.); kpolewska@gumed.edu.pl (K.P.); aleksanderoch@gumed.edu.pl (A.O.); bogdan.biedunkiewicz@gumed.edu.pl (B.B.); adeb@gumed.edu.pl (A.D.-Ś.); 2Department of Radiology, University Center for Maritime and Tropical Medicine, 81-519 Gdynia, Poland; anna.susmarska@gmail.com; 37th Naval Hospital in Gdańsk, 80-305 Gdańsk, Poland; e.puchalska@7szmw.pl (E.P.-R.); puchola@gmail.com (A.P.); k.szabat@7szmw.pl (K.S.); 4Department of Occupational, Metabolic and Internal Diseases, Faculty of Health Science, Medical University of Gdansk, 81-519 Gdynia, Poland; mrenke@gumed.edu.pl

**Keywords:** COVID-19, SARS-CoV-2, pneumonia, ACE inhibitors, vitamin D, chronic kidney disease, hemodialysis

## Abstract

The group most at risk of death due to COVID-19 are patients on maintenance hemodialysis (HD). The study aims to describe the clinical course of the early phase of SARS-CoV-2 infection and find predictors of the development of COVID-19 severe pneumonia in this population. This is a case series of HD nonvaccinated patients with COVID-19 stratified into mild pneumonia and severe pneumonia group according to the chest computed tomography (CT) pneumonia total severity score (TSS) on admission. Epidemiological, demographic, clinical, and laboratory data were obtained from hospital records. 85 HD patients with a mean age of 69.74 (13.19) years and dialysis vintage of 38 (14–84) months were included. On admission, 29.14% of patients had no symptoms, 70.59% reported fatigue followed by fever—44.71%, shortness of breath—40.0%, and cough—30.59%. 20% of the patients had finger oxygen saturation less than 90%. In 28.81% of patients, pulmonary parenchyma was involved in at least 25%. The factors associated with severe pneumonia include fever, low oxygen saturation and arterial partial pressure of oxygen, increased C-reactive protein and ferritin serum levels, low blood count of lymphocytes as well as chronic treatment with angiotensin converting enzyme inhibitors; while the chronic active vitamin D treatment was associated with mild pneumonia. In conclusion, even though nearly one-third of the patients were completely asymptomatic, while the remaining usually reported only single symptoms, a large percentage of them had extensive inflammatory changes at diagnosis with SARS-CoV-2 infection. We identified potential predictors of severe pneumonia, which might help individualize pharmacological treatment and improve clinical outcomes.

## 1. Introduction

The COVID-19 pandemic has been going on for more than two years, its death toll devastating healthcare systems around the world, and it had already led to the death of nearly 6 million of the world’s population by the end of February 2022. The global fatality rate is currently estimated in most countries to be between 1% and 2%. This changes over time and is dependent on many factors, including the number of tests performed (which helps identify more asymptomatic and mild cases), the dominant virus variant, increased rate of infection in younger people, improvements in health care management, and more recently, the vaccination rate of the population [1].

The group most at risk of death are patients with chronic kidney disease on maintenance hemodialysis (HD) [2]. The 28-day probability of death before commencing field vaccination was 25% for all patients, and 33.5% for subjects who were admitted into hospitals according to the European Renal Association COVID-19 Database (ERACODA) report [3]. In our previous study, we showed the extremely high mortality of COVID-19 HD patients from the North of Poland, with a fatality rate of up to 43.81% in subjects over 74 years old [4]. In addition, the survivors suffer from a persistent symptom complex called post-COVID-19 syndrome [5]. We demonstrated that as many as 81% of HD patients report that at least one symptom of COVID-19 persists after six months [6]. These symptoms are dominated by fatigue and shortness of breath resulting from permanent pulmonary architectural distortion and irreversible pulmonary dysfunction [7,8]. It translates into a reduction in the quality of life, which in this group of patients remains at a very low level [6,9].

Vaccines significantly reduced the risk of severe disease and mortality during the course of COVID-19 [10]. Today, however, we already know that breakthrough SARS-CoV-2 infections develop in some vaccinated HD patients. This is due to the weakening of vaccine immunity over time and the emergence of new, more infectious and immune-bypassing variants of the virus [11]. In addition, HD patients are characterized by impaired innate immunity and a worse humoral and cellular response to vaccines than in the general population [12,13,14]. Various methods of pharmacological treatment, including inhaled steroids, casirivimab-imdevimab, remdesivir, and tocilizumab are highly hoped for, which, as research shows, if applied early enough, can significantly improve the prognosis [15,16,17,18]. Therefore, the need to identify patients infected with SARS-CoV-2 and those who develop pneumonia, which may be fatal or have persistent sequelae, remains as urgent as possible. Studies conducted in the general population identified the following predictors for COVID-19 pneumonia: older age, obesity, higher fever at presentation, hypoxemia, laboratory markers of inflammation, and lymphopenia [19,20,21]. The aim of the study below is to describe the clinical course of the early phase of SARS-CoV-2 infection in HD patients and find predictors of the development of COVID-19 severe pneumonia in this population.

## 2. Materials and Methods

### 2.1. Study Design and Settings

This is a case series of HD patients with COVID-19 conducted at the 7th Naval Hospital in Gdańsk. Through a decision of the local health authorities, all HD patients from the Pomeranian Voivodeship with SARS-CoV-2 infection during the second wave of the pandemic were obligatorily hospitalized and hemodialyzed in a dedicated unit at the 7th Naval Hospital [4]. We included all individuals aged 18 years and older with laboratory confirmation of SARS-CoV-2 infection and available chest CT scan on admission, hospitalized between 6 October 2020 and 28 February 2021. The study patients were not vaccinated against COVID-19. The vaccination process for dialysis patients in Poland was only initiated in late January 2021. Stratification based on median chest computed tomography (CT) COVID-19 pneumonia total severity score (TSS) on admission divided the cohort into a severe changes group with extensive inflammatory changes (severe pneumonia) and a mild changes group with limited inflammatory changes (mild pneumonia).

### 2.2. Definitions

Laboratory confirmation for SARS-CoV-2 infection was defined as a positive result of the RT-PCR assay of a nasal or pharyngeal swab. COVID-19 pneumonia was diagnosed by the presence of opacity on chest CT in an amount exceeding 1% of the lungs and confirmed by a radiologist’s assessment. The Charlson comorbidity index (CCI) was calculated by summing the assigned weights of all comorbid conditions presented by the patients, according to the formula of Charlson et al., on admission [22]. The frailty index was calculated on a scale of 1–9, according to the Clinical Frailty Scale, and applies functional descriptors and pictographs. An index of 1 represents very fit, and 9 represents terminally ill [23].

### 2.3. Data Collection and Procedures

Epidemiological, demographic (age, sex), admission clinical and laboratory data, chest CT findings, and outcome were obtained from patients’ hospital records. All data were verified by two physicians (E.P.-R., A.P.). All CT scans performed in the first 24 h of hospitalization were evaluated by CT pneumonia analysis software by Siemens Healthineers providing automatic segmentation and quantification of lungs, lobes, and affected areas (volume and percentage) in the lung parenchyma like ground-glass opacities, consolidation, crazy paving pattern, etc. According to this scale, each of the 5 lung lobes was assessed for the percentage of lobar involvement. If the parenchymal involvement was 0, 1–5%, 5–25%, 25–49%, 50–75% and >75% they were assigned a score of 0, 1, 2, 3, 4 and 5 respectively. The TSS was reached by adding the 5 lobal scores to each other (range from 0 to 25) [24]. A specialist radiologist (A.S.) with long professional experience made additional assessments and possible corrections of the obtained results.

### 2.4. Statistical Analyses

Continuous measurements are presented as mean (SD, standard deviation) if they were normally distributed or median (IQR, interquartile range) if they were not, and categorical variables are presented by numbers and percentages. No imputation was made for missing data. In strata analyses of factors associated with severe COVID-19 pneumonia, the median TSS of 7 points was used as the cut-off value. Due to the small number of cases and exploratory character of the research, multivariate analyses were not performed. The Mann-Whitney U test, *t*-test, and Chi Square test were applied to analyze differences between groups according to the data type. All analyses were performed using the software Statistica 13. *p* < 0.05 was considered significant.

## 3. Results

### 3.1. Patients Characteristic

During the study period, 133 patients were hospitalized for COVID-19. Of these, we achieved access to complete admission data, along with a chest CT scan, records of 85 individuals who were finally included in the study. Among them, 52.94% were male, and 47.06% were female. The mean age of the cohort was 69.74 (13.19), and the median dialysis vintage was 38 (14–84) months. Overall, 62 (72.94%) were recovered and discharged from the hospital, while 23 (27.06%) died during their hospital stay. The mean time to discharge was 17.77 (7.37) days, while the median time to death was 10 (3–16) days. Table 1 and Appendix A present the characteristics of patients and their chronic home treatment.

### 3.2. Clinical Presentation on Admission

On admission, 29.14% of patients had no symptoms. In these patients, the diagnosis was accidental when they were tested for SARS-CoV-2 after prior contact with infected individuals. The most common presenting symptom in all of the cohort was fatigue (70.59%), followed by fever (44.71%), shortness of breath (40.0%), and cough (30.59%).

In the first 24h of presentation, mean (SD) finger oxygen saturation was 93.47 (5.62)%, body temperature was 36.92 (0.68) °C, and mean systolic blood pressure 140.76 (24.54) mmHg. Seventeen (20.0%) patients had a finger oxygen saturation of less than 90%. Laboratory investigation revealed a mean WBC count of 6.13 (2.87) with a median (IQR) lymphocyte count of 0.87 (0.63–1.27). Forty-nine (57.65%) patients were lymphopenic. All patients had showed raised serum C-reactive protein and D-dimers with a median level of 53 (13.6–117.4) mg/L, and 1161.8 (685.89–1842.2) ng/mL, respectively. The mean arterial partial pressure O_2_ was 67.04 (22.94) mmHg. Twelve (14.13%) patients presented with arterial partial pressure O_2_ value below 60 mmHg on admission. Detailed findings on admission are presented in Table 2.

### 3.3. Imaging Evaluation

Among 85 patients included in the study, 66 (77.65%) had CT evidenced COVID-19 pneumonia on admission. The predominant chest CT features in patients with confirmed pneumonia included ground-glass opacities 63 (74.12%), crazy paving pattern 53 (62.35%), consolidation 35 (41.18%), pleural effusion 34 (40%), and linear opacities 24 (28.24%). Typical radiographic changes observed in our patients are shown in Figure 1. These changes had mainly a bilateral distribution—61 (92.42%) in lower lobes. In 25 (37.88%) and 22 (33.33%) patients, pulmonary parenchyma was involved in 1–5% and 5–25%, respectively. In 14 (21.21%) and 5 (7.6%) patients, pulmonary parenchyma was involved in 25–50% and >50%, respectively. The median TSS in all cohorts was 7 (4–11) (Table 3).

### 3.4. Factors Associated with Severe COVID-19 Pneumonia

The factors associated with extensive inflammatory changes in lungs (severe pneumonia) at diagnosis COVID-19 include chronic treatment with angiotensin converting enzyme inhibitors (ACEI) (*p* = 0.02), while chronic treatment with active vitamin D was associated with limited changes (mild pneumonia) (*p* < 0.011) (Table 1). The clinical features on admission of fever (*p* = 0.01), low finger oxygen saturation (*p* < 0.001), low arterial partial pressure of oxygen (*p* = 0.02), increased indicators of inflammation including serum C-reactive protein (*p* < 0.001) and ferritin (*p* = 0.03), and low count of lymphocytes (*p* < 0.001) were associated with severe pneumonia expressed by a higher TSS score. Detailed results are presented in Table 1 and Table 2.

## 4. Discussion

In this study, we retrospectively analyzed the clinical presentation of HD patients with newly diagnosed SARS-CoV-2 infection. The most common early clinical symptoms at admission included fatigue, fever, dyspnea, and cough and did not differ from those seen in other studies in HD patients [25,26,27]. At the time of diagnosis, nearly one-third of the patients were completely asymptomatic, while the remaining subjects usually reported single symptoms much less frequently than in the general population, which may be due to numerous immune disturbances and impaired reactivity [28]. At the same time, as many as 20% of patients presented with reduced blood oxygenation requiring urgent oxygen therapy. Similar to other reports, more than half of the cases were lymphopenic and frequently had increased markers of inflammation [26,27,29,30].

Consistently in several previous reports in COVID-19 individuals from the general population, our results showed that COVID-19 pneumonia in HD patients features predominant ground glass opacities, mainly bilateral, with lower lung zones being mostly involved [31,32]. Ground-glass opacities refer to the area of increased attenuation in the lung on computed tomography with preserved bronchial and vascular markings and can be taken as an indicator of the early stage of pneumonia [33]. A crazy paving pattern, defined as a linear pattern superimposed on GGO resembling irregularly shaped paving stones, was observed in 62% of the cohort and 82% in the group with a severe course. This is important given the fact that this appearance can be considered as an indicator of disease progression [34]. In the general population, these changes are observed in a much smaller percentage ranging from 5 to 36% [35]. Area of the increased lung opacity with obscuration of underlying bronchovascular markings refers to consolidation, which is also an indicator of disease progression and appears in the general population at a later stage of the disease [36]. This mixed pattern, showing a perilobular and peripheral distribution, suggests the presence of a secondary organizing pneumonia [33]. In our cohort, it was observed in over 40% of patients already on admission. Importantly, all these changes were observed in nearly 78% of all patients diagnosed with SARS-CoV-2 infection, including those who did not report any symptoms. The remaining cases also presented opacities, which, however, did not exceed 1% of lung parenchyma and were not treated as pneumonia for this study. In the study of Turgutalp et al. in a Turkish HD population, this percentage was 89.6% [27]. These results indicate that the chest CT may be a valuable method for diagnosing COVID-19 in HD patients. Fang et al. found that the sensitivity of chest CT for COVID-19 diagnosis was even greater than RT-PCR (98% versus 71%) [37]. In nearly 30% of our patients, the changes at the time of diagnosis covered more than 25% of the lung parenchyma. All this data indicates a frequent occurrence of serious lung lesions yet at the time of diagnosis and a very rapid course of the inflammatory process in HD patients, closely corresponding to the extremely high mortality rate in this group of patients. Of note, the mean time of hospital admission was 2.2 days from the onset of symptoms, and the median time to death in the severe course group was 4 days after admission. The pleural effusion usually not seen in COVID-19 turned out to be a common (40%) CT chest finding similar to the study of Turgutalp et al. [27]. It may be related to the specificity of dialysis patients, more specifically to fluid overload and concomitant heart failure.

Identifying patients with rapidly developing pneumonia and a high risk of developing acute respiratory distress syndrome (ARDS) as soon as possible might help individualize pharmacological treatment and optimal utilization of medical resources. It was demonstrated that the sooner pharmacological treatment is started in such patients, the greater are the chances of survival [15,16,17,18]. We found that patients with more extensive lung changes on admission objectified by a higher TSS were likely to present with fever, had decreased finger oxygen saturation and arterial oxygen partial pressure, increased laboratory indicators of inflammation including serum c-reactive protein and ferritin, and decreased lymphocytes count. The predictive value of inflammation level and lymphopenia concerning the progression of COVID-19 and a poor outcome was reported in previous studies [20,38]. In our recent study, high CRP and D-dimer levels upon admission were strongly associated with a 3-month mortality risk in HD patients [39]. Other studies also show a predictive value of blood oxygenation indices for the severity of pneumonia and mortality of COVID-19 patients [21,40,41].

An important part of the study was the assessment of potentially modifiable factors that could affect the patient outcome. Chronic home treatment might affect susceptibility and prognosis. In particular, it was postulated that ACE inhibitors could act as a potential risk factor for SARS-CoV-2 infection and poor outcome by upregulating ACE2, a viral entry co-receptor for the virus [42,43]. The results of our study seem to confirm this hypothesis. Patients who received chronic ACEI treatment were more likely to have more extensive lung lesions. However, there is also enough evidence that allows stating the opposite hypothesis [42]. Firstly, there are limited findings showing changes in serum or pulmonary ACE2 levels after ACEI. Secondly, ACE2 and angiotensin (1–7) have been found to be protective in several different acute lung injury models [44]. For instance, a large population-based study revealed an association of ACE inhibitors with lower COVID 19 incidence. Variations between different ethnic groups observed in this study raise the possibility of ethnic-specific effects of ACE inhibitors on COVID-19 disease susceptibility and severity, which deserves further study [45]. The relationship between the use of ACEI and the incidence and severity of COVID-19 should be investigated in further studies, especially in vaccinated HD patients. Determination of ACE2 expression could help confirm such a relationship and our hypothesis [46].

Low 25-hydroxy vitamin-D (25-OH D) level is associated with proinflammatory cytokines levels and was demonstrated to be an independent predictor of COVID-19 severity in the general population [47]. However, research into vitamin D supplementation in the prevention and treatment of COVID-19 provides inconsistent results [48,49]. In the recent study, we originally found that in HD patients who were treated with active vitamin D, there may be a lower risk of 3 month death from COVID-19 [39]. There was substantial clinical and methodological heterogeneity of the conducted studies, mainly because of different supplementation strategies, formulations, vitamin D status of participants, and reported outcomes [50]. Our cases did not have determined 25-OH-D levels. Following the KDIGO guidelines, some of them received 1-alpha-hydroxyvitamin D3 (Alfacalcidol), aimed to normalize calcium and phosphorus levels and maintain parathormone within two to nine-times the normal upper limit [51]. Such treatment turned out to be a significant factor associated with less extensive inflammation in the lungs, thus supporting the hypothesis of a beneficial effect of vitamin D on the prognosis of patients. It may be based on the anti-inflammatory effect of active vitamin D and the prevention of a cytokine storm in COVID-19. Very recent data suggest that proinflammatory interleukin-6 functions may be redirected to the production of anti-inflammatory interleukin-10 by vitamin D in activated human helper T cells [52]. Vitamin D also plays an important role in the innate and acquired defense against infections [53]. The active form of vitamin D3 was shown to upregulate the production of antimicrobial peptides, i.e., cathelicidin LL-37 in macrophages and lymphocytes involved in the process of autophagy, i.e., the intracellular killing of pathogens in infected cells [54]. Vitamin D is also required for acquired immunity and antimicrobial activity mediating by Th1 and Th2 cells [55]. Finally, it may promote the stabilization of the endothelium and the barrier function in the presence of inflammatory mediators [56]. Following this lead, numerous preclinical studies have demonstrated that vitamin D suppresses the replication of Mycobacterium tuberculosis in vitro, which may translate to preventing and developing the disease [53]. The hypothetical antibacterial effect of vitamin D may also contribute to its potential protective effect on the extent of lung lesions in the course of SARS-CoV-2 pneumonia. Multiple studies reported an unexpectedly high incidence of multidrug-resistant gram-negative, gram-positive bacteria, and fungi infections among patients with COVID-19 admitted to the intensive care unit [57].

The question remains whether the potentially beneficial effects of vitamin D in HD patients apply only to its active formulations or also to its native form, i.e., cholecalciferol and ergocalciferol. A nonrandomized trial of 158 patients on hemodialysis demonstrated reduced inflammation markers, serum-intact parathyroid hormone, and erythropoietin-stimulating agent dose after 6 months of cholecalciferol supplementation [58]. On the contrary, other research that has measured inflammation in patients on hemodialysis after vitamin D supplementation has not found reductions in cytokines or T-cell or monocyte mediators [59,60]. It is known that more than half of the patients on hemodialysis are deficient in total serum 25-hydroxy vitamin D [25 (OH) D], and administration of native vitamin D corrects these levels without significant increases in serum calcium or phosphorus. On the other hand, impaired 1-α hydroxylation in the kidney and non-fully investigated extrarenal vitamin activation raises doubts about the effectiveness of nutritional forms of vitamin D in this patient group [61].

The study, to our best knowledge, is the first to analyze potential predictors of COVID-19 pneumonia in HD patients. The study’s strengths include representing the entire spectrum of the disease, from asymptomatic to severe cases, and detailed quantitative artificial intelligence-assisted assessment of the extent of inflammatory changes in the lungs. A limitation of the study is its observational design, which only permits the description of associations. Secondly, the relatively small sample of cases with CT chest images allows drawing conclusions of an exploratory character only.

## 5. Conclusions

Despite the fact that nearly one-third of the patients were completely asymptomatic, with the remaining usually reporting only single symptoms, a large percentage of them had extensive inflammatory changes at diagnosis with SARS-CoV-2 infection. Fever, elevated markers of inflammation, and decreased blood oxygenation are predictors of the extent of the inflammatory changes in the lungs expressed by a high TSS score in CT. Chronic treatment with ACE inhibitors may increase the risk, while the use of active vitamin D may reduce the risk of developing severe pneumonia. The potential risks of ACEI therapy in the context of SAR-CoV-2 infection and the well-known cardioprotective properties of ACEI need to be weighed. Until more substantial data is available to define recommendations, chronic treatment of heart failure with ACEI in dialysis patients should not be interrupted. The use of active vitamin D in dialysis patients may improve their prognosis during the COVID-19 pandemic. However, it should be carried out in accordance with the current KDIGO recommendations [51], taking into account the levels of calcium, phosphorus, and the concentration of the parathyroid hormone.

## Figures and Tables

**Figure 1 viruses-14-00451-f001:**
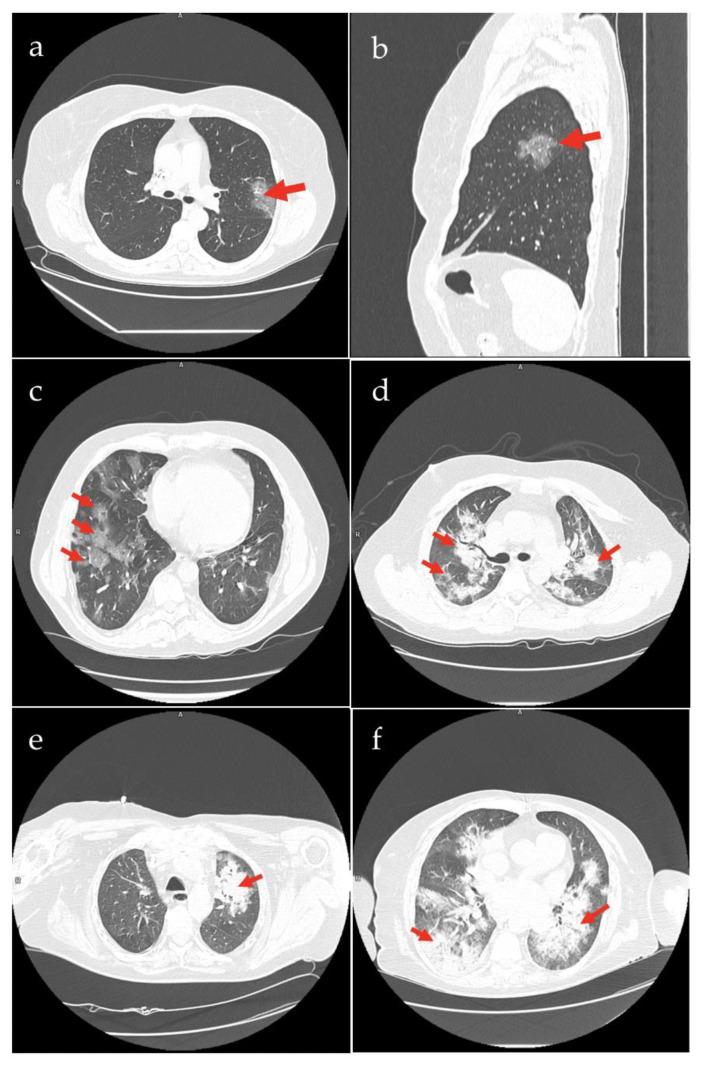
Representative chest CT images with pathological changes (red arrows) from our cases: (**a**,**b**). ground-glass opacity in the upper lobe of the left lung; (**c**). multifocal changes of ground-glass opacity with a crazy-paving pattern symptom; (**d**). peribronchial, bilateral opacities of organizing pneumonia with air bronchogram (**e**). unilateral consolidations; (**f**). bilateral consolidations.

**Table 1 viruses-14-00451-t001:** Characteristics of cases stratified according to the extent of inflammatory lesions in the lungs.

Variable	ALL	Mild	Severe	*p* Value *
		Changes	Changes	
N	85	47	38	
Men n (%)	45 (52.94)	27 (57.45)	18 (47.37)	0.35
Age, years	69.74 (13.19)	70.23 (14.63)	69.13 (11.33)	0.71
Body mass index, kg/m^2^	25.34 (4.79)	25.21 (4.73)	25.49 (4.92)	0.79
Blood group n (%)				
A	36 (42.35)	17 (36.17)	19 (50.0)	0.19
B	14 (16.47)	9 (19.15)	5 (13.16)	0.46
AB	6 (7.05)	2 (4.25)	4 (10.53)	0.26
O	29 (34.12)	19 (40.43)	10 (26.31)	0.17
Dialysis vintages, months median (IQR)	38 (14–84)	33 (14–85)	46 (21–82)	0.74
Past kidney transplantation n (%)	7 (8.23)	5 (10.64)	2 (5.26)	0.37
Hemodialysis dose per week, hours	11.63 (1.59)	11.50 (1.69)	11.8 (1.47)	0.39
Hemodialysis access n (%)				
AVF/AVG	29 (34.12)	19 (40.42)	10 (26.31)	0.17
Dialysis catheter	56 (65.88)	28 (59.57)	28 (73.68)	0.17
Charlson comorbidity index	7.64 (2.23)	7.49 (2.22)	7.82 (2.26)	0.51
Fragility index	4.16 (1.36)	4.02 (1.34)	4.34 (1.38)	0.28
Comorbidities n (%)				
Arterial hypertension	83 (97.65)	47 (100)	36 (94.74)	0.44
Diabetes	45 (52.94)	24 (51.06)	21 (55.26)	0.69
Ischemic heart disease	34 (40.0)	18 (38.30)	16 (42.10)	0.72
Congestive heart failure	31 (36.47)	14 (29.79)	17 (44.74)	0.15
Malignancy	7 (8.23)	5 (10.64)	2 (5.26)	0.37
Chronic pulmonary disease	4 (4.71)	3 (6.38)	1 (2.63)	0.42
**Chronic drug treatment n (%)**				
Beta-blockers	58 (68.23)	32 (68.08)	26 (68.42)	0.97
Active oral vitamin D	57 (67.05)	37 (78.72)	20 (52.53)	0.011
Statins	36 (42.35)	19 (40.42)	17 (44.74)	0.69
Calcium channel blockers	27 (31.76)	14 (29.79)	13 (34.21)	0.66
LMWH between HD session days	20 (23.53)	12 (25.53)	8 (21.05)	0.63
ACEI	17 (20.00)	5 (10.64)	12 (31.58)	0.02
ARBs	3 (3.53)	2 (4.25)	1 (2.63)	0.69
Oral anticoagulant	5 (5.88)	3 (6.38)	2 (5.26)	0.83
Cinacalcet	3 (3.53)	1 (2.12)	2 (5.26)	0.44
LMWH during dialysis	54 (63.53)	28 (59.57)	26 (68.42)	0.39
UFH during dialysis	31 (36.47)	19 (40.43)	12 (31.58)	0.39

Legend: AVF, arteriovenous fistula; AVG, arteriovenous graft; ACEI, angiotensin-converting enzyme inhibitors; ARB, angiotensin II receptor blocker; LMWH, low molecular weight heparin; UFH, unfractionated heparin; HD, hemodialysis. *—mild changes group vs. severe changes group. Data are presented as mean (SD) unless otherwise indicated.

**Table 2 viruses-14-00451-t002:** Clinical presentation on admission to the hospital.

Variable	ALL	Mild Changes	Severe Changes	*p* Value *
N Asymptomatic course n (%) Symptomatic course n (%) Fatigue Fever Shortness of breath Cough Chills Diarrhea Appetite loss Smell/taste loss Headache Joint pains Rhinorrhea Myalgia Sore throat Sleep disturbances Symptoms duration days	85 25 (29.41) 60 (70.59) 38 (44.71) 34 (40.00) 27 (31.76) 26 (30.59) 12 (14.12) 12 (14.12) 6 (7.06) 2 (2.35) 4 (4.71) 4 (4.71) 2 (2.35) 3 (3.53) 2 (2.35) 1 (1.18) 2.2 (1.41)	47 14 (29.79) 33 (70.21) 22 (46.81) 13 (27.66) 14 (29.79) 12 (25.53) 7 (14.89) 6 (12.77) 3 (6.38) 1 (2.13) 4 (8.51) 2 (4.25) 2 (4.25) 2 (4.25) 0 (0.0) 1 (2.13) 2.51 (3.58)	38 11 (29.95) 27 (71.05) 16 (42.10) 21 (55.26) 13 (34.21) 14 (36.84) 5 (13.16) 6 (15.79) 3 (7.89) 1 (2.63) 0 (0.0) 2 (5.26) 0 (0.0) 1 (2.63) 2 (5.26) 0 (0.0) 1.8 (2.61)	0.93 0.93 0.66 0.01 0.66 0.26 0.82 0.69 0.79 0.88 0.25 0.83 0.87 0.87 0.44 0.88 0.82
**Observations on admission** Body temperature, °C Body temperature > 37.5 °C n (%) Finger oxygen saturation, % Finger oxygen saturation < 90% n (%) Systolic blood pressure, mmHg Diastolic blood pressure, mmHg Heart rate, beats/min	36.92 (0.68) 17 (20) 93.47 (5.62) 17 (20) 140.76 (24.54) 79.27 (13.21) 80.81 (14.12)	36.85 (0.61) 8 (17.02) 95.44 (3.14) 4 (8.5) 141.91 (23.86) 80.3 (13.12) 81.81 (14.33)	37.0 (0.76) 9 (23.68) 90.87 (7.03) 13 (34.21) 139 (25.6) 77.97 (13.38) 79.74 (13.96)	0.31 0.44 <0.001 0.003 0.64 0.43 0.56
**Laboratory values on admission** Haemoglobin (g/dL; nr: 13.0–17.5) Leucocytes (×10^9^ per L; nr: 3.5–9.5) Neutrophils (×10^9^ per L; nr: 1.8–6.3) ** Lymphocytes (×10^9^ per L; nr: 1.1–3.2) ** Lymphocytes < 1.1 × 10^9^ Platelets (×10^9^ per L; nr: 125.0–350.0) C-reactive protein (mg/L; nr: 0.0–5.0) ** Serum ferritin (ng/mL; nr: 21.0–274.7) ** ALAT (U/L; nr: 9.0–50.0) Prothrombin time (s; nr: 10.5–13.5) APTT (s; nr: 21.0–37.0) D-dimer (ng/mL; nr: 0–500) ** Arterial pO_2_, mmHg Arterial pO_2_ < 60 mmHg Arterial pCO_2_, mmHg	10.75 (1.75) 6.13 (2.87) 3.72 (2.65–6.03) 0.87 (0.63–1.27) 49 (57.65) 192 (69.84) 53 (13.60–117.40) 1039.0 (678.2–2022.0) 21 (11–29) 13.58 (5.62) 43.03 (13.57) 1161.8 (685.89–1842.2) 67.04 (22.94) 12 (14.12) 34.13 (6.77)	10.81 (1.71) 5.8 (2.69) 3.22 (2.38–5.06) 1.09 (0.76–1.46) 22 (46.81) 187.91 (64.68) 23.55 (6.9–95.0) 820.7 (680.45–1319) 22 (12–28) 13.35 (5.26) 42.98 (15.88) 946.65 (712.8–1501.4) 75.68 (15.95) 2 (4.25) 34.16 (4.69)	10.66 (1.81) 6.52 (3.07) 4.26 (2.9–6.79) 0.72 (0.57–1.02) 27 (71.05) 197.08 (75.96) 93.4 (53–176.7) 1820 (678.2–2300) 20.5 (11–30) 13.87 (6.13) 43.09 (10.39) 1257.25 (646.5–1845.4) 60.99 (25.42) 10 (26.31) 34.11 (8.1)	0.26 0.94 0.94 0.001 0.025 0.56 <0.001 0.03 0.31 0.72 0.97 0.64 0.02 0.004 0.98

Legend: nr, normal range; ALAT, alanine aminotransferase; APTT, activated partial thromboplastin time; pCO_2_, partial pressure of carbon dioxide; pO_2_, partial pressure of oxygen. Data are presented as mean (SD) or unless otherwise indicated. **—median (IQR) *—mild changes group vs. severe changes group.

**Table 3 viruses-14-00451-t003:** Chest computed tomography features.

Variable	ALL	Mild Changes	Severe Changes	*p* Value *
N **Type of lungs changes n (%)** Unilateral infiltration Bilateral infiltration Ground-glass opacity (GGO) Consolidation Air bronchogram Linear opacities Crazy-paving pattern Pleural effusion **Involved lung lobes n (%)** Upper lobes Middle lobe Lower lobes	85 12 (14.12) 68 (80) 63 (74.12) 35 (41.18) 14 (16.47) 24 (28.24) 53 (62.35) 34 (40) 74 (87.06) 55 (65.7) 80 (94.18)	47 11 (23.40) 31 (65.96) 25 (53.19) 4 (8.5) 7 (14.89) 12 (25.53) 22 (46.81) 12 (25.53) 36 (76.60) 20 (42.55) 42 (89.36)	38 1 (2.63) 37 (97.37) 38 (100) 31 (81.58) 7 (18.42) 12 (31.58) 31 (81.56) 22 (57.89) 38 (100) 35 (92.1) 38 (100)	0.006 <0.001 <0.001 <0.001 0.66 0.001 0.001 0.002 0.006 <0.001 0.15
**Degree of lung involvement **** Total opacities—% lungs GGO—% lungs Consolidation—% lungs Total severity score (TSS)—points	5 (1–22) 4 (1–17) 1 (3–5) 7 (4–11)	1 (0–4) 1 (0–3) 0 (0–1) 5 (2–6)	24 (10–38) 20 (10–31) 3 (1–9) 14 (10–16)	<0.001 <0.001 <0.001 <0.001

Legend: GGO, ground-glass opacities; *—mild changes group vs. severe changes group; **—median (IQR).

## Data Availability

Detailed data are available on request from the corresponding author.

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
