# Peer review of "Angiotensin Converting Enzyme Inhibitors May Increase While Active Vitamin D May Decrease the Risk of Severe Pneumonia in SARS-CoV-2 Infected Patients with Chronic Kidney Disease on Maintenance Hemodialysis"

_viruses, 2022, doi:10.3390/v14030451_

Round 1

Reviewer 1 Report

I read with great interest the paper. I find it well wrote and well presented. Knowledge on SARS CoV2 is ongoing and share experience from different countries can help to increase SARS CoV2 knowledge.  Below my suggestions

  1. Introduction: updata data on SARS CoV2 wordwilde at the time of resubmission.
  2. Methods and results: are well presented and data on your patients are interesting
  3. Discussion: discuss better the role of Vitamin D. Different studies and this review ( see and cite Patti et al Potential Role of Vitamins A, B, C, D and E in TB Treatment and Prevention: A Narrative Review. Antibiotics (Basel). 2021 Nov 5;10(11):1354. doi: 10.3390/antibiotics10111354) suggest how strong is relation between vitamin (especially Vitamin D) and infectious diseases and lung diseases. Furthermore, add the role also of antimicrobial resistance as compliance for the patients during hospitalization to SARS CoV2 pneumonia (see and cite Segala FV, Bavaro DF, Di Gennaro F, Salvati F, Marotta C, Saracino A, Murri R, Fantoni M. Impact of SARS-CoV-2 Epidemic on Antimicrobial Resistance: A Literature Review. Viruses. 2021 Oct 20;13(11):2110. doi: 10.3390/v13112110.)
  4. Conclusion: give some public health suggestion that came from your well wrote paper

Author Response

REVIEWER 1:

1.Introduction: updata data on SARS CoV2 wordwilde at the time of resubmission.

It was done as requested (line 44)

2.Discussion: discuss better the role of Vitamin D. Different studies and this review ( see and cite Patti et al Potential Role of Vitamins A, B, C, D and E in TB Treatment and Prevention: A Narrative Review. Antibiotics (Basel). 2021 Nov 5;10(11):1354. doi: 10.3390/antibiotics10111354) suggest how strong is relation between vitamin (especially Vitamin D) and infectious diseases and lung diseases. Furthermore, add the role also of antimicrobial resistance as compliance for the patients during hospitalization to SARS CoV2 pneumonia (see and cite Segala FV, Bavaro DF, Di Gennaro F, Salvati F, Marotta C, Saracino A, Murri R, Fantoni M. Impact of SARS-CoV-2 Epidemic on Antimicrobial Resistance: A Literature Review. Viruses. 2021 Oct 20;13(11):2110. doi: 10.3390/v13112110.)

It was done as requested (line  288-302).

3. Conclusion: give some public health suggestion that came from your well wrote paper

It was done as requested (line  330-337)

Reviewer 2 Report

Remarks to the author:

In this study the authors investigated the clinical presentation SARS-CoV-2 infection and predictive factors for the development of COVID-19 severe pneumonia in patients on maintenance hemodialysis. Epidemiological, demographic, clinical and laboratory data were obtained from hospital records. Total of 85 patients were included into the study and the factors associated with severe pneumonia were identified as fever, low oxygen saturation and arterial partial pressure of oxygen, increased C-reactive protein and ferritin serum levels, low blood count of lymphocytes, chronic treatment with angiotensin converting enzyme inhibitors whereas the chronic active vitamin D treatment was associated with the mild pneumonia. Thus the authors suggests that chronic treatment with ACE inhibitors may increase the risk, while the use of active vitamin D may reduce the risk of developing severe pneumonia.

Specific comments:

  1. Revise the statement for the aim in the abstract and the introduction to convey the same idea.
  2. The predictive factors for the development of severe pneumonia in other most common chronic diseases mentioned in literature needs to be discussed briefly in the introduction.
  3. An additional experiment to measure the ACE2 expression in those patients would confirm the hypothesis of the authors that “ACE inhibitors could act as a potential risk factor for SARS-CoV-2 infection and poor outcome by up-regulating ACE2, a viral entry co-receptor for the virus”
  4. Figure 1- Labelling for Right and Left sides in the images and referring to lesion precisely would aid the reader to identify lesions specifically.

Thanks.

Author Response

REVIEWER 2:

1.Revise the statement for the aim in the abstract and the introduction to convey the same idea.

It was done as requested (line 22 and 77).

2. The predictive factors for the development of severe pneumonia in other most common chronic diseases mentioned in literature needs to be discussed briefly in the introduction.

It was done as requested (line 75-77).

3. An additional experiment to measure the ACE2 expression in those patients would confirm the hypothesis of the authors that “ACE inhibitors could act as a potential risk factor for SARS-CoV-2 infection and poor outcome by up-regulating ACE2, a viral entry co-receptor for the virus”.

Relevant commentary and reference were added in the discussion as requested (line  269-270)

4. Figure 1- Labelling for Right and Left sides in the images and referring to lesion precisely would aid the reader to identify lesions specifically.

Pathological changes are marked with red arrows in each photo of Figure 1, which allows for their precise identification.